# Evaluating Germplasm of Cultivated Oat Species from the VIR Collection under the Russian Northwest Conditions

**DOI:** 10.3390/plants11233280

**Published:** 2022-11-28

**Authors:** Vitaliy S. Popov, Valentina I. Khoreva, Alexei V. Konarev, Tatyana V. Shelenga, Elena V. Blinova, Leonid L. Malyshev, Igor G. Loskutov

**Affiliations:** 1N.I. Vavilov All-Russian Institute of Plant Genetic Resources, 42-44 Bolshaya Morskaya Street, 190000 St. Petersburg, Russia; 2Department of Agrochemistry, Faculty of Biology, Saint Petersburg State University, 7-9 Universitetskaya Emb., 190000 St. Petersburg, Russia

**Keywords:** Nikolai Vavilov, the VIR collection of plant genetic resources, cultivated *Avena* L. species, nutritional value, useful agronomic characters

## Abstract

Oat is one of the most widespread and important cereal crops in the global agricultural production. Searching for new high-yielding and nutritious forms continues to be relevant, especially under the global trend of climate change, when most local oat cultivars may become economically inefficient. Spring oat accessions from VIR collection served as the material for this study; their origin is diverse, as they came from 11 countries. The basic nutritional value (the content of protein, oil, starch, and *β*-glucans) and characters important for breeding (plant height, panicle length, number of spikelets, number of grains per panicle, 1000 grain weight, and grain yield) were analyzed in 49 accessions of the cultivated covered oat species: *Avena sativa* L., *A. strigosa* Schreb., *A. abyssinica* Hochst., and *A. byzantina* Coch., grown under the conditions of the Russian Northwest (Leningrad Province) for two years. Variability parameters, interspecific and intervarietal differences, and the effect of weather conditions were assessed. Sources of useful agronomic traits were identified; they can be used to expand the range of the source material for the development of new high-yielding and highly nutritious oat cultivars adapted to local cultivation conditions. It is demonstrated that the VIR collection has a great potential for contemporary food and feed production and for the breeding of new oat cultivars for various purposes. Thus, the contribution of Nikolai Vavilov to the plant genetic resources investigation for the benefit of humanity is invaluable.

## 1. Introduction

Oat is one of the most widespread and important cereal crops in the system of agricultural production in temperate climates. This crop is important for food and feed industries due to the chemical composition of its grain. Oat-based food products are distinguished for their high protein content with a favorable amino acid composition [1]; high content of oil, rich in fat-soluble vitamin E and polyunsaturated fatty acids, with an optimal fatty acid composition [2,3,4,5,6]; high content of oat-specific antioxidants (avenanthramides) [7]; and a sufficient set of vitamins and macro- and micronutrients [6,8,9], and an optimal ratio of amylose and amylopectin in starch [10,11,12]. Special attention has been paid in recent years to oat and barley *β*-glucans, with their numerous functional and bioactive properties. Beta-glucans belong to the class of indigestible polysaccharides found in grains, yeasts, bacteria, algae, and fungi [13,14].

Oat-based foods are beneficial for people with insulin-resistant forms of diabetes, hypertension, obesity, and dyslipidemia [15]. In addition, the safety of including oat in a gluten-free diet has been confirmed in long-term studies [15,16,17]. Thus, oat is a highly nutritious cereal crop suitable for curative, preventive, and functional nutrition [1,2,3,4,5,6,7,8,9,10,11,12,13,14,15,16,17]. In addition to food and feed nutritive value, there are such important breeding-oriented characters, such as plant height (PH), panicle length (LP), number of spikelets (NSP), number of grains per panicle (NG), 1000 grain weight (W1000), and grain yield, or seed productivity (SP). These descriptors characterize the economically important features [18]. Thus, it is important to undertake integrated evaluations of useful agronomic characters as well as food and feed values.

With the global climate change, many of the local crop cultivars, including oats, may become economically inefficient. Therefore, searching for new forms of high-yielding and nutritious oat cultivars is now especially pertinent. The optimal approach to solving this problem is a systematic and integrated study of the biodiversity preserved in the collections of plant genetic resources. The works started by Vavilov and continued by his colleagues on the germplasm material from the VIR collection are still underway [19]. Since Vavilov’s times, the VIR global collection of plant genetic resources, including *Avena* L., has been replenished by collecting missions sent to different regions of the 81 world and through germplasm exchange with other plant genetic resources centers and 82 research institutes. Currently, the collection of oats includes about 14,000 accessions, representing various species and originating from different parts of the world.

Screening numerous accessions from the VIR collection includes the assessment of their main breeding-oriented characters and biochemical indicators to identify plant forms/accessions, which offer the best combinations of traits for breeding. Such selected accessions, according to the guidelines [20], are called “donors or sources of useful agronomic traits”, which entails their in-depth studying and subsequent utilization in various breeding programs. The present research effort is one of the stages in an integrated study of accessions representing cultivated oat species from the VIR collection.

The objective was to assess the variability of the various cultivated oat accessions for the following biochemical and agronomic characters: protein, oil, starch and *β*-glucan content, PH, LP, NSP, NG, W1000, and SP. The assayed accessions represent different ploidy level: *A. sativa* (2 n = 42), *A. byzantina* (2 n = 42), *A. abyssinica* (2 n = 28) and *A. strigosa* (2 n = 14) from the VIR collection. The trials were conducted for two years in the fields of the Russian Northwest region (Leningrad Province). Specifically, the variability of all studied indicators and properties, interspecific and intervarietal differences, and the effect of weather conditions on the studied characters were observed and documented. A number of oat accessions were identified as sources of high agronomic value for various uses: plant breeding, fodder production, and food industry, including the use for medicinal and functional products.

## 2. Results and Discussion

### 2.1. Description of Characters Valuable for Breeding

More than 250 plant characters showing hereditary differences [20,21,22,23,24] have been identified by now within the oat species. Information on the characters valuable for breeding in the studied oat accessions—plant height (PH), panicle length (LP), number of spikelets per panicle (NSP), number of grains per panicle (NG), 1000 grain weight (W1000), and grain yield, or seed productivity (SP)—are presented in Appendix A in the Appendix A.

Average SPs (g/m^2^) for *A. sativa*, *A. abyssinica*, *A. byzantina* and *A. strigosa* accessions were 276.3, 55.0, 203.3 and 96.7 in 2014, and 350.0, 191.7, 307.3 and 106.0 in 2015, respectively. Grain yield of *A. sativa* accessions in 2014 was higher compared to the reference cultivar ‘Privet’ (260.0), while in 2015 there were no accessions that exceeded the SP value of the reference (490.0). However, the accessions of *A. sativa* and *A. byzantina* had the highest average grain yields, because their plants were the most vigorous. Grain yields of all oat cultivars in 2015, which was more humid and cool during the grain ripening period, were higher compared to 2014.

According to the W1000 value, the oat accession could be with small grains (W1000 is less than 26 g), large grains (W1000 is more than 35 g) and 128 middle ones [25]. *A. sativa* and *A. byzantine* accessions, taken in current investigation, belonged to the large grains group, *A. abyssinica* and *A. strigosa* to the small grains. This division was confirmed by the conclusions of N.A. Rodionova et al. [25]: *A. sativa* and *A. byzantine* accessions, by virtue of their polyploidy, belong to the group of genotypes with large grains, while *A. abyssinica* and *A. strigosa* accessions belong to the group with small grains. The variability range of W1000 values in 2014 and 2015 was nearly the same as was from 18.9 (k-15438, Ethiopia, *A. abyssinica*) to 51.5 g (k-15484, Brazil) in 2014, and from 18.1 (k-15436, Ethiopia, *A. abyssinica*) to 52.6 g (k-15484, Brazil) in 2015. The highest mean of W1000 values corresponded to *A. byzantina*, and then followed accessions of *A. sativa*, *A. strigosa* and *A. abyssinica* (41.8, 41.0, 21.5, 20.4 in 2014, and 44.0, 42.3, 23.1, 17.8 g in 2015, respectively).

NG in an oat panicle can vary from less than 30 pcs. up to more than 60 pcs. [25]. In the studied oat accessions, it ranged from 23 (k-15489, Brazil) to 75 (k-15381, Belarus) in 2014, and from 26 (k-15489, Brazil) to 79 (k-15381, Belarus) in 2015. The average of NG in all accessions of the studied species in 2015 was higher compared to 2014. The highest NG was identified in *A. abyssinica* accessions, then followed *A. sativa*, *A. strigosa* and *A. byzantine.*

NSP in the studied set of accessions varied from 15 (k-15435, Ethiopia, *A. abyssinica*) to 60 (k-15478, Brazil) in 2014, and from 13 (k-15489, Brazil) to 55 (k-15437, Ethiopia, *A. abyssinica*) in 2015. The average values of NSPs in the studied species did not differ significantly in both years of study. Due to the large panicle length, the accessions of *A. strigosa* and *A. abyssinica* demonstrated the highest NSP, while *A. byzantina* and *A. sativa* accessions had the smallest ones (52.7, 43.7, 31.7, 26.4 in 2014, and 48.0, 46.7, 34.7, 26.9 in 2015, respectively).

Oat LP parameters can range from less than 15 to more than 30 cm [25]. In the current investigation, the LP variation was from 13 (k-15489, Brazil) to 26 (k-15437, Ethiopia, *A. abyssinica*) in 2014, and from 12.5 (k-15489, Brazil) to 27 (k-15437, Ethiopia, *A. abyssinica*) in 2015. Accessions of the studied species had approximately the same average LPs in both years of study. The highest value was detected for the *A. abyssinica* accessions, then followed *A. strigose, A. sativa* and *A. byzantine* (24.0, 20.3, 19.7, 18.1 in 2014 and 27.3, 20.5, 18.3, 17.4 in 2015, respectively).

Rodionova et al. [25] approved that oat PH can vary in the range from more than 170 to less than 60 cm. Among accessions taken in the current study, the PH values varied from 80 (k-15489, Brazil) to 150 (k-15478, k-15479, Brazil) in 2014, and from 55 (k-15489, Brazil) to 165 cm (k-15478, Brazil) in 2015. The average values of PH in 2014 were slightly higher than that in 2015. *A. strigosa* accessions demonstrated the highest PHs, which is typical for this species [25], and those of *A. byzantine* demonstrated the smallest values. *A. sativa*, *A. abyssinica* occupied an intermediate position (148.3, 114.0, 126.7, 116.7 in 2014, and 146.7, 98.3, 104.3, 116.7 cm in 2015, respectively). 

Field experiments carried out in the central highlands of Ethiopia revealed the relationship between species affiliation of studied accessions and manifestation of their agronomic traits [26,27], which is confirmed by our results.

The ranking of the oat species according to the agronomic descriptions in ascending order was as follows: (PH) *A. byzantina*, *A. sativa*, *A. abyssinica*, *A. strigose*; (NSP) *A. byzantina*, *A. sativa*, *A. abyssinica*, *A. strigosa*; (SP) *A. strigosa*, *A. abyssinica*, *A. byzantina*, *A. sativa*; (LP) *A. strigosa*, *A. abyssinica*, *A. sativa*, *A. byzantina*; (NG) *A. byzantina*, *A. strigosa*, *A. sativa*, *A. abyssinica*; (W1000) *A. abyssinica*, *A. strigosa, A. sativa*, *A. byzantina*. The assessed oat species had almost similar orders of arrangement according to LP and NSP. The values of the NSP indicator in *A. byzantina* accessions were twice lower than in *A. strigosa* accessions. Panicles of *A. byzantina* accessions were, on average, almost 1.5 times shorter than those of *A. abyssinica*. Average W1000 values turned out to be almost the same in both years of research. The values of the LP and NSP indicators in 2015 did not differ much from those in 2014. Average PH values were higher in 2014 compared to 2015. The highest NG and SP values were recorded for oat accessions grown in 2015. Thus, the grain yield and plant height indicators in oat accessions were inversely related to each other.

According to Grib and Kadyrov, the shortening of stems entails a decrease in some plant productivity parameters (NSP, W1000 and SP) [22]. Thus, we distinguished those accessions that exceeded the productivity of the reference cv. ‘Privet’ (33.0 pcs., 38.1 g, and 375.0 g/m^2^). Five accessions (*A. sativa*: k-15443, 15444, 15457; *A. byzantine:* k-15465) demonstrated higher values of SP, three (*A. sativa*: k-15457, 15473, *A. byzantine:* k-15465)—NSP, four (*A. sativa*: k-15444, k-15453, k-15457 and k-15473)—W1000 compare with cv. ‘Privet’ (Appendix A). The selected oat accessions included ones (k-15443, k-15444, k-15453, k-15473 and k-15465) with lower PH (107.5, 107.5, 110.0, 105.0, 110.0 cm, respectively) and k-15457 with higher (127.5 cm) compare with PH of cv. ‘Privet’ (112.5 cm).

The range of PH of 48 oat’s genotypes conducted in the Indian Agricultural Research Institute, Wellington, was narrower (from 61.47 to 118.63 cm) and PH mean lower (94.03 cm) [28] compared with the VIR accessions.

According to Premkumar et al., the range of oat’s W1000 (from 21.86 to 58.45) was wider, mean of W1000 (39.00 g) higher [29], but range of NG (32.00-74.33) was narrower and mean of NG (51.47) higher [28] compared with the data obtained in the current investigation. 

### 2.2. Description of Biochemical Indicators

All accessions selected for the study demonstrated significant variability in protein, oil, starch and *β*-glucan content (Table 1, Figure 1A–D).

The highest scope of variation in protein content was observed in the accessions of *A. sativa* and *A. byzantina* in 2014 (9.0% and 6.7%, respectively). The scope of variation for *A. abyssinica* and *A. strigosa* was 2.1 and 2.2%, respectively. In 2015, the maximum scope variability was manifested by the accessions of *A. byzantina* (4.7%) and *A. sativa* (3.3%). Varietal differences were statistically less significant in *A. abyssinica* (0.3) and *A. strigosa* (1.1%) accessions. Two-year average varietal differences in protein content for *A. sativa*, *A. abyssinica*, *A. strigosa* and *A. byzantina* were 13.7–12.2, 15.6–15.0, 19.5–17.8 and 14.7–12.2%, respectively. Thus, different cultivated oat species can have variable protein content (Table 1, Figure 1A). 

Protein content in oat grain is one of the most important indicators of quality; this parameter is associated with the nutritional and fodder qualities of the crop [30]. On average, the amount of protein ranges from 11.7 to 15.1% [31] and depends on the cultivar, weather conditions, and the level of agricultural practice. In our studies, 2015 was wetter and less favorable for protein accumulation and grain formation, as shown by the lower protein content in all species compared to 2014 (Figure 4, Appendix A). Persistently high protein content over the two-year period of studies was found in diploid accessions of *A. strigosa*: k-15478, k-15479 and k-15480 from Brazil (19.5–17.8%); tetraploid *A. abyssinica*: k-15436 from Ethiopia (15.2–16.6%); hexaploid *A. sativa*: k-15476 and k-15483 from the USA and Brazil (18.3–14.7%) and *A. byzantina*: k-15474 from the USA, and k-15484 and k-15488 from Brazil (17.9–14.5%). The values of cv. ‘Privet’ (k-14787, reference) were significantly lower (12.5–11.8%) (Appendix A).

According to published data [32], the amount of protein in the grain of crops grown in the northern regions of the Russian Federation increased with the movement of their cultivation areas from north to south and from west to east. Hence, it can be assumed that protein content in the same oat cultivars that were grown in Leningrad Province will be higher if they are cultivated farther south or east. Our data are compatible with the results obtained by Kazakh, Polish and Italian researchers [10,33,34].

The nutritional and biological value of oats also depends on the oil content; its levels vary from 3.5 to 6.2% in covered oat cultivars of different origin, and from 7.1 to 9.0% in naked oat cultivars [12]. The grain of some cultivars, however, may contain up to 18% of oil [2]. If the breeding of high-oil oat cultivars proves successful, then in the future oats can be classified as oilseeds and efficiently used in chemical and pharmaceutical industries [35].

Among the studied oat accessions, the highest varietal variability in oil content was observed in *A. byzantina* accessions: 3.1 to 5.8% in 2014, and 3.2 to 6.1% in 2015. Slightly less variable in oil were *A. sativa* accessions: 3.3 to 5.7% in 2014, and 3.7 to 6.0% in 2015. These accessions demonstrated varietal differences of 2.7 and 2.9% in 2014, and 2.4 and 2.3% in 2015, respectively. The minimum varietal variability was observed in diploid *A. strigosa* accessions: 3.9 to 4.1% in 2014, and 3.9 to 4.3% in 2015, as well as in tetraploid *A. abyssinica* accessions: 4.4 to 4.5% in 2014, but in 2015 the amount of oil increased by 0.8–0.7% to 5.2–5.4% (Table 1, Figure 1B). In 2015, a large amount of rainfall in July contributed to the formation of moisture reserves in the soil and provided an opportunity for intensive oil synthesis in oat grains, despite the dry weather in August (Figure 4, Appendix A). These results demonstrated that the decisive factor in the accumulation of oil in oat grains was the presence of moisture at the stages of grain development and ripening. On average, over two years, the differences in oil content for *A. sativa* accessions were 4.4–4.6, for *A. byzantina* 4.4–4.7, for *A. abyssinica* 4.5–5.2, and for *A. strigosa* 4.0–4.1%. Accessions of *A. strigosa* had the lowest oil content (Figure 1B, Appendix A). With such an insignificant variability in most of the studied accessions, 10 sources of high oil content under the conditions of Leningrad Province were identified: for *A. abyssinica*, k-15438 from Ethiopia (5.4); for *A. sativa*, k-15453, k-15455 from Russia and k-15483 from Brazil (5.48, 5.21 and 5.86, respectively); for *A. byzantina*, k-15368 from Portugal, k-15397 from the UK, k-15435 from Ethiopia, k-15470 from Germany, k-15475 from the USA, and k-15489 from Brazil (5.54, 5.66, 5.24, 5.18, 5,63 and 5.52, respectively). The oil content in their grains was higher than in the reference (4.95%) (Appendix A). According to published data, the variability of oil content in covered oat cultivars had close values in the range of 4.4 to 7.2% [31,34,35]. 

The oat grain by its starch structure resembles the rice grain but differs significantly from the wheat grain [10]. Oat starch decomposes slowly but completely, without provoking an abrupt rise in blood glucose levels after eating, which is beneficial for patients with diabetes. Slow absorption of glucose into the blood is believed to be caused by the presence of a significant number of *β*-glucans in oat [36]. Starch content is a relatively stable character, with a lesser degree of dependence on growing conditions compared to protein content [37]. Our research confirmed this fact: protein content in all studied oat cultivars was on average higher in 2014 than in 2015. Meanwhile, starch content did not demonstrate such a clear dependence, and the effect of the year on accessions of different oat species manifested itself in different ways. According to published data [38], starch content in the oat grain ranged from 23.7 to 69.5%, depending on the species and cultivar. The starch content in the oat accessions we studied was lower and ranged from 27.3 to 42.75% in 2014, and from 30.72 to 45.52% in 2015 (Table 1, Figure 1C). The significant sorts variability of protein and starch content in *A. sativa, A. byzantina* and *A. strigose* in the different year of reproduction was detected in the current study and associated with the peculiarities of the response of those species to changing environmental conditions (temperature; rainfall), which we noted earlier [21].

The study of covered oat accessions performed in 2010–2012 in the environments of Western Siberia (southern forest-steppe) demonstrated that the variation of starch content was within the range from 38.6 to 48.8%, with an average of 43.8% [39], which was somewhat higher than our data. *Avena sativa* accessions grown in the Netherlands on sandy and clay soils demonstrated an even higher starch level in the grain: 45.6 to 66.5%, on average, 53.1%, which is also higher than in the accessions we studied [40]. 

Oat, along with barley, is also characterized by high contents of non-starch polysaccharides, *β*-glucans, and glucose polymers. The content of *β*-glucans in the oat grain is, on average, lower than in barley (from 3 to 11), but higher than in rye (1–2%), wheat (<1%), and other cereals, where they are present in trace amounts [15]. Commercial samples of oat bran contain 7–10% of *β*-glucans [41]. Cereals are potentially the least expensive source of glucans, which favors their use as food additives and food ingredients in functional foods available to a wide range of consumers. In addition, there are no data on any negative effects after eating foods rich in *β*-glucans from oat or barley flour or their extracts [42]. At the same time, the problem of reducing the amount of dietary fiber exhibiting antinutritional properties is an urgent problem in animal feed production [15]. 

Our data demonstrated that the amplitude of intervarietal variability of *β*-glucans in the 308 studied accessions was 2.19 to 5.77% in 2014, and 2.46 to 4.83% in 2015. The least variability in both years of research was observed in *A. byzantina*: 0.03 and 0.07% (Table 1, Figure 1D). The high and stable content of *β*-glucans was preserved in the accessions of *A. strigosa*: k-15479 from Brazil (3.73), *A. sativa*: k-15447, 15451, 15452 and 15454 from Russia (4.42, 4.54, 3.74 and 4.08, respectively), and *A. byzantina*: k-15368 from Portugal (4.26), compared to cv. ‘Privet’ (reference) (2.95%). These accessions are the most valuable for use in food industry with the purpose of expanding the assortment of functional foods. As a result of the study, accessions with the lowest content of *β*-glucans (2.49–2.46%) were identified: k-15480 (*A. strigosa*) from Brazil (2.49%), and k-15436 (*A. abyssinica*) from Ethiopia (2.46%); they can be used as forage in animal husbandry (Appendix A). 

The results of studying *β*-glucans in oat grain, presented by different authors, differ significantly from one another. According to V.I. Polonskiy et al., naked oat accessions did not noticeably prevail over covered oats in the content of *β*-glucans. The grain of covered oat accessions could accumulate from 2.9 to 5.2% of *β*-glucans [31]. At the same time, recent studies have demonstrated that the content of *β*-glucans in covered oats is about 3% [43]. The study of 1700 mutagenic lines of the Swedish oat cultivar ‘Belinda’ resulted in finding that their content of *β*-glucans ranged from 1.8 to 7.5% [15]. Our data were fully consistent with the published information: the average value of *β*-glucans was 3.52% in 2014, and 3.30% in 2015.

Environmental conditions were observed to have a significant impact on the content of *β*-glucans in oat grain during the development of the endosperm: dry and warm weather contributes to a significant increase in *β*-glucans [15]. Our data confirmed those findings: the smallest amount of *β*-glucans was in the grain harvested in 2015. Thus, high humidity during the ripening phase is less favorable for the synthesis of *β*-glucans (Figure 4, Appendix A). 

Thus, the protein content in the studied material decreased across the accessions in the following order: *A. strigosa*, *A. abyssinica*, *A. byzantina* and *A. sativa*, while the oil content drops as follows: *A. abyssinica*, *A. byzantina*, *A. sativa* and *A. strigosa*. Higher protein levels were recorded for accessions of all species in 2014, and oils in 2015. *A. sativa* accessions had the highest starch values, followed in descending order by *A. byzantina*, *A. strigosa* and *A. abyssinica* accessions. The effect of weather conditions on starch accumulation was not as obvious as it was with oil and protein content. *A. strigosa* and *A. byzantina* accessions had the highest starch content in 2014, and *A. byzantina* and *A. sativa* in 2015. The content of *β*-glucans increases in the accessions in the following order: *A. abyssinica*, *A. strigosa*, *A. byzantina* and *A. sativa*. For all species, except *A. abyssinica*, the content of *β*-glucans was higher in 2014: dry and warm weather was more favorable for the synthesis of these compounds. It can be concluded that dry and warm weather contributed to protein accumulation in the Leningrad Province, while wet and cooler weather, on the contrary, stimulated oil accumulation. This fact is consistent with published data. Higher humidity sets up favorable conditions for the work of enzymes involved in the biosynthesis of oil. Under moisture deficiency, transpiration grows faster than the accumulation of carbohydrates in leaves. The result is a decrease in oil content and an increase in protein formation [23,44]. 

A successful breeding normally requires the understanding of the interactions among grain quality indicators (protein, starch, oil and *β*-glucan content) for specific crop growing conditions. Thus, we performed the analysis of trait correlation, and the results are presented in Table 2.

Correlations between the main indicators of grain quality confirmed the impact of weather conditions on the strength and direction of these relationships. In 2014, there were weak positive correlations between protein and oil and between starch and oil, and a moderately strong positive correlation between starch and protein accumulation (Table 2). In 2015, the correlation between protein and oil content remained weakly positive, while the correlations between protein and starch and between oil and starch changed to negative (−0.61 and −0.24, respectively). There were different levels of an inverse relationship between the content of *β*-glucans and other quality indicators, with the exception of a positive correlation between the content of *β*-glucans and starch in 2015 (Table 2). 

According to published sources [23,29], the amount of carbohydrates, which are the substrate for oil formation, should decrease with an increased protein synthesis. Therefore, the reduction of oil content in the grain can be significant. This explains the direction of correlations between the content of protein and oil and between protein and starch [29], which is confirmed by our correlation data for the dry 2014 and the humid 2015 (Table 2).

B.P. Pleshkov demonstrated that under moisture deficiency, at elevated temperatures, and with high nitrogen content in the soil, the metabolism in plants shifted towards an increase in protein synthesis. Under reverse conditions, the process of carbohydrate and starch synthesis would predominate in plant seeds [23]. This fact was confirmed by the correlations we obtained for 2014 and 2015. 

Other researchers found a high positive correlation between the levels of oil and *β*-glucans in the grain of oat cultivars in different years of cultivation. The authors suggested that oil concentration could be an indirect indicator of the content of *β*-glucans in the grain, despite the fact that these biochemical components had different accumulation mechanisms [31,43]. We found a negative correlation between these indicators.

### 2.3. Statistical Processing of the Results Obtained

The biochemical data collected from this work were further statistically processed using the classical discriminant analysis, supplemented by the canonical analysis and identification of canonical variables (Figure 2). The resulting model includes all studied features: protein, oil, starch and dry matter content. The most informative parameter was “protein content” (correlates with canonical axis 1), which makes it possible to differentiate accessions belonging to the *A. strigosa* species with 100% accuracy. “Oil content” and “starch content” (canonical axis 2) allowed us to differentiate accessions of *A. byzantina* and *A. sativa* (45.0 and 78.2% of correct solutions, respectively). “Dry weight” (canonical axis 3) did not affect the distribution of an accession (0% of correct solutions). Accessions of *A. abyssinica* did not stand out on the basis of this set of observations, occupying an intermediate position. 

It was established that the main biochemical parameters of oat accessions were affected by specific assayed species, by the weather conditions in the year of their cultivation, and by the level of breeding improvement among the cultivars in the studied set, associated with the year of entry into the collection and expressed through the VIR catalogue number (individual number of each accession). This confirms the dependence of biochemical parameters both on the species-specific and individual features of the studied genotypes as well as on weather conditions (air temperature, and rainfall). All calculations are presented in Appendix A. 

The interplay between biochemical indicators of grain quality and useful agronomic characters is shown in Table 3.

All characteristics were closely interrelated: the yield of oat grain negatively correlated with the size of the plant organs, which may be explained by an increase in the number of grain husks due to the elongation of the stem and panicle. Grain yield in 2014 positively correlated with 1000 grain weight; the yield in 2015 positively correlated with the number of grains per panicle. Protein content had a negative correlation with 1000 grain weight, number of grains, and yield in 2014, but positively correlated with oat plant height, panicle length, and number of spikelets in 2015. There was a weak negative correlation between oil content and the numbers of spikelets and grains per panicle. Correlations of starch content with other indicators in 2014 and 2015 had opposite dependences. The strongest correlations were found between starch content and number of grains and between starch content and yield. 

R. Premkumar et al. [28] revealed an inverse relationship between the height of oat plants and the weight of 1000 grains (−0.295), as in our study, but the relationship was weaker than the one calculated in our experiment (−0.29 in 2014, and −0.69 in 2015). The same authors established a direct relationship between the height of oat plants and the number of grains per panicle (−0.052) [28]. Our research yielded the opposite results (0.44 in 2014, and 0.7 in 2015); we also identified a negative correlation between the number of grains per panicle and the weight of 1000 grains (−0.65 in 2014, and −0.70 in 2015), contrary to the findings of Premkumar et al. (0.633) [28]. 

Variability of the studied characters in different years was demonstrated using factor analysis; its structure is presented in Table 4 and Figure 3A,B. 

In 2014, the first factor determined the linear dimensions of the plant: plant height and panicle length (0.785), and partially, 1000 grain weight (−0.551); the second, 1000 grain weight (0.713), grain yield (0.904), and protein content (−0.719); the third, starch content (0.892). In 2015, the first factor retained its structure, while the second was associated with changes in indicators: grain yield (−0.764), starch content (−0.772), and protein content (0.706). The effect of the third factor on the variability of oat characters in 2015 requires further confirmation.

The correlation structure and factor analysis revealed complex interactions among individual oat characters and a strong effect of the weather conditions in the year of cultivation (Figure 3A,B).

Only *A. strigosa* and *A. abyssinica* accessions, due to their low level of ploidy, were clearly isolated, and in 2015, their differentiation was more obvious.

Our statistical analysis also demonstrated a close interplay among all the studied indicators and reliably confirmed the effect of the species, individual features of the cultivar, and the weather specificity in the cultivation area (Russian Northwest) on useful agronomic characters and on nutritional value. These findings allowed for the identification of sources of valuable agronomic traits that can be used to expand the range of feed and food (functional and medicinal) production as well as to develop new high-yielding and highly nutritious oat cultivars adapted to local conditions.

The unevenness of the accession’s set of the current study is obvious. There is a predominance of widespread *A. sativa* and *A. byzantina* accessions and a relatively small number of sparsely distributed *A. abyssinica* and *A. strigosa*. Such a ratio is due to the insertion of *A. abyssinica* and *A. strigosa* accessions in the current experiment as oat forms that are atypical for this region of agricultural production (North-West of the Russian Federation). Thus, our findings regarding *A. abyssinica* and *A. strigosa* are largely preliminary. However, our effort made it possible for a larger experiment to include *A. abyssinica* and *A. strigose* accessions and to identify *A. abyssinica* and *A. strigose* forms valuable for the breeding process, including interspecific crossing. 

## 3. Material and Methods

### 3.1. Research Materials

The materials for this study were spring oat accessions of various geographic origin (from 11 countries) with useful agronomic traits, recently included into the VIR collection (Table 5 and Appendix A).

All the studied accessions belong to cultivated oat species: *Avena sativa*, *A. byzantina*, *A. abyssinica*. and *A. strigosa*, and are represented by covered oat genotypes.

The selected set of oats were grown under the conditions of the Russian Northwest (Pushkin SIB (59°71′ n.l., 30°38′ w.l.), Leningrad region). Sowing operations, observations and harvesting were performed in 2014–2015 according to the guidelines for the study and conservation of the global collection of barley and oats [24] at Pushkin and Pavlovsk Laboratories of VIR. The reference, cv. ‘Privet’ (k-14787), was commercialized and approved for large-scale agricultural production for the northwestern region of the Russian Federation. Sowing time was optimal for this region. The area of the plots was 1 m^2^. Two plots per accession. The reference cultivar was planted after every 20 plots in the sowing scheme.

The soils in the experimental field were soddy, slightly podzolic sandy loams, with neutral acidity (pH 7.1–7.6). The depth of the humus horizon was up to 23 cm, and the humus content was 2.1–3.0%. The reserve of mobile potassium forms was medium, and that of phosphorus was high. The climate in this agroclimatic zone is moderately warm, and there are cool summers in some years. The warmest month of the year is July, with a mean long-term air temperature of 16.5–17.7 °C. The total number of positive temperatures during the growing period (from seeding stage to harvest) is 2100–2300 °C. The temperatures above 10 °C persist for 105–115 days. The amount of rainfall during the growing season is 550–600 mm/year.

The weather conditions during the years of research differed: the mean temperatures and humidity for June, July and August in 2014 and 2015 are presented in Appendix A and in Figure 4A,B. As far as rainfall is concerned, 2015 was more humid and less warm. 

### 3.2. Assessment of Useful Agronomic Characters

The field study of each oat cultivar in the context of its useful agronomic characters: plant height (PH), panicle length (LP), number of spikelets (NSP), number of grains (NG), 1000 grain weight (W1000), and grain yield (SP), was carried out in accordance with the guidelines for the study and conservation of the global collection of barley and oats [24] at the stage of full ripeness.

### 3.3. Biochemical Analysis 

Each accession was represented by 50 g of grains mix. Prior to the analysis, the grains of each oat accession were ground into flour using a CM 290 Cemotec laboratory disc mill (FOSS, Sweden). The biochemical analysis was performed at the Department of Biochemistry and Molecular Biology of VIR applying VIR’s methods [45]. Each analysis was performed in two replications, and the obtained average values were statistically analyzed. Values were expressed in “% dry weight”.

Protein content was measured by the Kjeldahl method: 300 mg of the mixture (in 3–4 repetitions) was mineralized with 5 mL of concentrated sulfuric acid at 420 °C for 1.5 h. Nitrogen was determined using a Kjeltec 2200 semi-automatic analyzer (FOSS, Hillerød, Denmark) with an automatic distillation unit, followed by titration with a 0.1 N sulfuric acid solution. The total protein content was calculated from the nitrogen content using a factor of 5.7. Oil content was calculated by weighing the dry fat-free residue. The analysis was carried out in a Soxhlet apparatus, with petroleum ether used as a solvent (boiling point 40–70 °C). Starch content was measured by the Ewers polarimetric method. Two grams of the sample were hydrolyzed in 25 mL of 1% hydrochloric acid solution (in 50 mL volumetric flasks) in a boiling water bath (Lauda Hydro H 20 SW, LAUDA, Marlton, NJ, USA) for 15 min, and cooled to room temperature; then, 2.5 mL of phosphotungstic acid was added to precipitate polysaccharides, proteins, etc., and distilled water was added to a volume of 250 mL. The extract after filtering was poured into a polarizing cuvette 10 cm long and the rotation angle was measured on an SAC-I automatic polarimeter/saccharimeter (Atago, Japan, Tokyo). The conversion factor was 181.3. Beta-glucans were determined gravimetrically [46]. To inactivate *β*-glucanase and remove free sugars, certain lipids, proteins, etc., oatmeal was pretreated with 50% ethanol. The release of *β*-glucans from the aleurone layer of flour endosperm was carried out with 5% NaOH solution, and the final extraction was performed with 70% ethanol. Beta-glucans floated to the surface in the form of a bundle of fibers, which were then dried at a temperature of 100–102 °C to constant weight and weighed. The content of *β*-glucans was calculated on the basis of the product’s dry weight (%). 

### 3.4. Statistical Analysis

Statistical data processing: correlation analysis, classical discriminant analysis, supplemented by canonical analysis, and identification of canonical variables, factor analysis, was carried out using the Statistica 12.0 software package (StatSoft, Inc., Tulsa, OK, USA). (2019), STATISTICA (data analysis software system), version 12. www.statsoft.com) (accessed on 2 May 2022).

## 4. Conclusions

This study has generated a set of findings on different cultivated oat species under the Russian Northwest conditions. The obtained data helped to identify the sources of valuable agronomic traits, which can be used to develop new high-yielding and highly nutritious cultivars adapted to local cultivation conditions. The value of *A. strigosa* and *A. abyssinica* accessions in a wide range of traits was also demonstrated for the first time.

## Figures and Tables

**Figure 1 plants-11-03280-f001:**
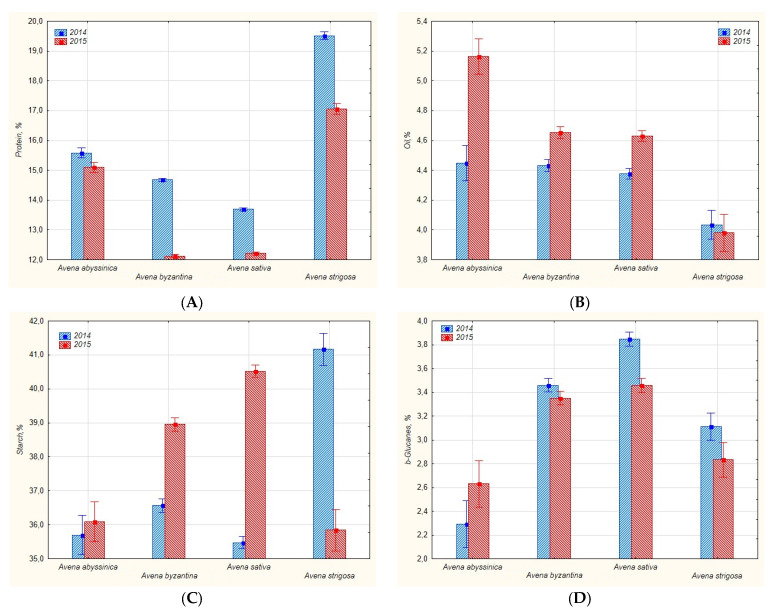
Protein (**A**), oil (**B**), starch (**C**) and *β*-glucan (**D**) content in *Avena* L. grain accessions from the VIR collection in 2014–2015.

**Figure 2 plants-11-03280-f002:**
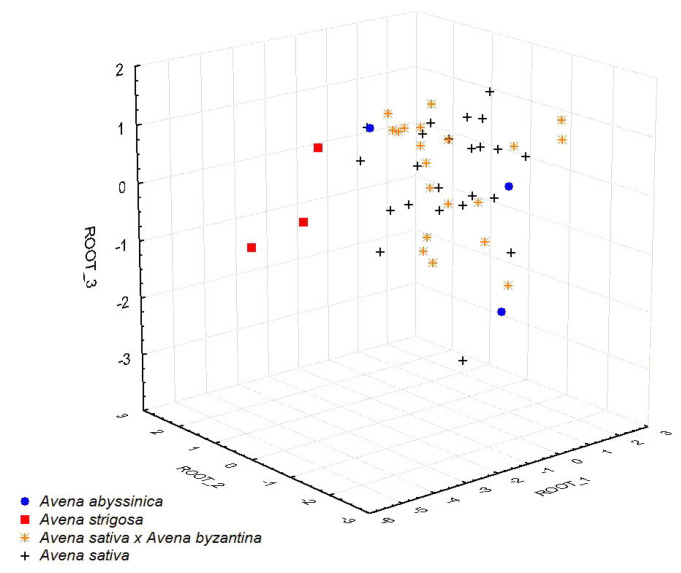
Distribution of the grain of *Avena* L. species accessions from the VIR collection in the space of three canonical axes.

**Figure 3 plants-11-03280-f003:**
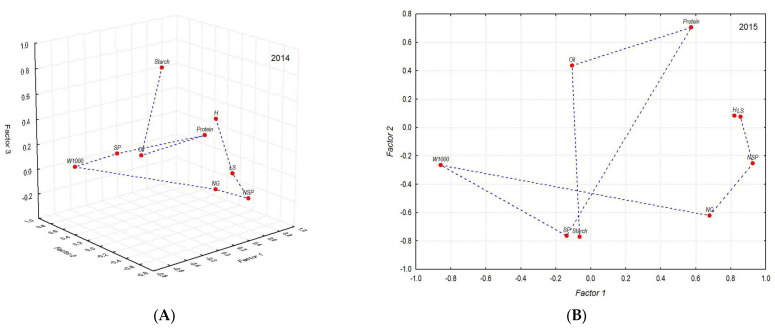
Factor structure of the variation of the studied indicators in 2014 (**A**) and 2015 (**B**).

**Figure 4 plants-11-03280-f004:**
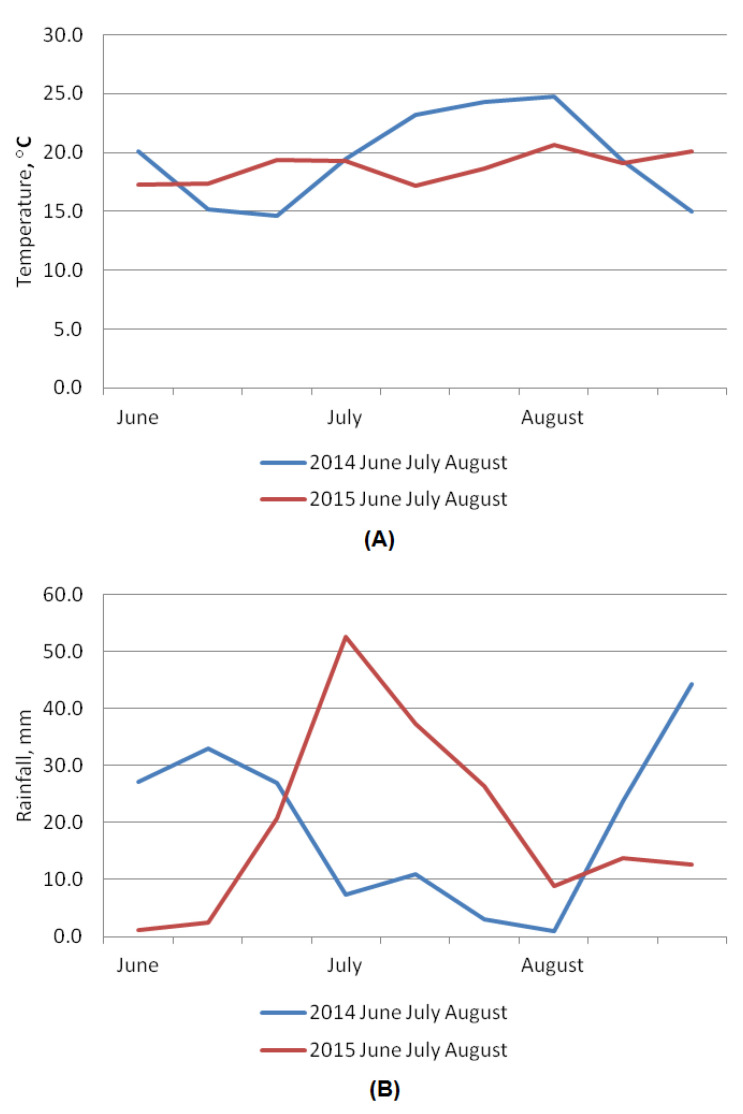
Diagrams of weather conditions: air temperature (**A**) and rainfall (**B**) in 2014–2015.

**Table 1 plants-11-03280-t001:** Average values of protein, starch, oil and *β*-glucan content in *Avena* L. grain accessions from the VIR collection, in % DW (dry weight).

*Avena* spp.	Years of Study	Number of Entries	Protein	Oil	Starch	*β*-Glucans
*A. sativa* L.	2014	23	13.65 ± 1.79	4.40 ± 0.69	35.58 ± 3.17	3.78 ± 0.83
	2015	23	12.20 ± 1.04	4.64 ± 0.70	40.45 ± 3.10	3.41 ± 0.56
Mean for 2 years			12.92	4.52	38.02	3.60
*A. abyssinica* Hochst.	2014	3	15.58 ± 1.22	4.45 ± 0.02	35.70 ± 1.84	2.29 ± 0.06
	2015	3	15.03 ± 0.17	5.24 ± 0.18	35.26 ± 3.47	2.63 ± 0.11
Mean for 2 years			15.31	4.84	35.48	2.46
*A. strigosa* Schreb.	2014	3	19.51 ± 0.98	4.03 ± 0.16	41.16 ± 1.67	3.11 ± 0.91
	2015	3	17.78 ± 0.57	4.14 ± 0.19	33.36 ± 1.45	2.84 ± 0.38
Mean for 2 years			18.65	4.09	37.26	2.97
*A. byzantina* Coch.	2014	20	14.65 ± 1.92	4.42 ± 0.75	36.36 ± 3.35	3.46 ± 0.26
	2015	20	12.20 ± 1.15	4.70 ± 0.79	38.68 ± 4.48	3.35 ± 0.58
Mean for 2 years			13.42	4.56	37.52	3.40
*A. sativa* L. (reference)	2014	1	12.51 ± 0.39	4.91 ± 0.03	37.88 ± 0.22	3.03 ± 0.11
	2015	1	11.80 ± 0.28	4.99 ± 0.04	38.90 ± 0.04	2.87 ± 0.00
Mean for 2 years			12.15	4.95	38.39	2.95

**Table 2 plants-11-03280-t002:** Correlation coefficients among the protein, starch, oil and *β*-glucan values in the grain of the *Avena* L. species accessions from the VIR collection in 2014–2015.

Variable	Protein *		Oil *		Starch *		*β*-Glucans **	
	2014	2015	2014	2015	2014	2015	2014	2015
**Protein ***	–	–	0.08	0.14	**0.42**	**−0.61**	**−0.53**	**−0.55**
**Oil ***	0.08	0.14	–	–	0.15	**−0.23**	**−0.38**	−0.21
**Starch ***	**0.42**	**−0.61**	0.15	**−0.23**	–	–	**−0.58**	**0.35**
***β*-Glucans ****	**−0.53**	**−0.55**	**−0.38**	−0.21	**−0.58**	**0.35**	–	–

* (n-184); ** (n-108).

**Table 3 plants-11-03280-t003:** Correlation coefficients between useful agronomic characters and biochemical indicators in the grain of *Avena* L. species accessions from the VIR collection in 2014–2015.

Indicators	PH *	LP *	NSP *	NG *	W1000 *	SP *	Protein **	Oil **	Starch **
2014									
**LP ***	**0.57**								
**NSP ***	**0.44**	**0.58**							
**NG ***	**0.41**	**0.62**	**0.61**						
**W1000 ***	**−0.29**	**−0.49**	**−0.65**	**−0.34**					
**SP ***	0.17	−0.03	−0.16	0.24	**0.50**				
**Protein ****	0.01	−0.14	0.10	**−0.32**	**−0.35**	**-0.57**			
**Oil ****	−0.20	−0.05	**−0.28**	**−0.35**	0.05	−0.05	0.08		
**Starch ****	0.03	−0.18	−0.21	−0.22	0.01	−0.12	**0.42**	0.15	
**2015**									
**LP ***	**0.64**								
**NSP ***	**0.70**	**0.69**							
**NG ***	**0.39**	**0.55**	**0.83**						
**W1000 ***	**−0.62**	**−0.75**	**−0.70**	**−0.39**					
**SP ***	−0.23	−0.21	0.06	**0.41**	0.26				
**Protein ****	**0.53**	**0.43**	**0.39**	−0.07	**−0.66**	**−0.51**			
**Oil ****	−0.13	0.10	−0.23	−0.20	−0.04	−0.19	0.14		
**Starch ****	−0.01	−0.06	0.08	**0.32**	0.24	**0.37**	**−0.61**	−0.23	

* useful agronomic characters: PH—plant height, cm; LP—panicle length, cm; NSP—number of spikelets per panicle, pcs; NG—number of grains per panicle, pcs; W1000—1000 grain weight, g; SP—grain yield (seed productivity), g/m^2^; ** biochemical indicators: protein, oil and starch content.

**Table 4 plants-11-03280-t004:** Factor structure for the characters of *Avena* L. accessions from the VIR collection in 2014–2015.

Indicators	2014			2015	
	Factor 1	Factor 2	Factor 3	Factor 1	Factor 2
**PH ***	**0.785**	0.134	0.322	**0.822**	0.084
**LP ***	**0.824**	−0.080	−0.088	**0.857**	0.076
**NSP ***	**0.776**	−0.371	−0.223	**0.925**	−0.252
**NG ***	**0.811**	0.162	−0.267	**0.680**	**−0.622**
**W1000 ***	**−0.551**	**0.713**	0.038	**−0.859**	−0.265
**SP ***	0.129	**0.904**	−0.004	−0.137	**−0.764**
**Protein ****	−0.108	**−0.719**	0.494	**0.574**	**0.706**
**Oil ****	−0.332	−0.054	0.244	−0.108	0.437
**Starch ****	−0.094	−0.077	**0.892**	−0.064	**−0.772**
**Factor share**	33.4	22.7	14.8	42.6	26.7

* useful agronomic characters: PH—plant height, cm; LP—panicle length, mm; NSP—number of spikelets per panicle, pcs; NG—number of grains per panicle, pcs; W1000—1000 grain weight, g; SP—grain yield (seed productivity), g/m^2^; ** biochemical indicators: protein, oil and starch content.

**Table 5 plants-11-03280-t005:** List of accessions of cultivated *Avena* L. species from the VIR collection used as the research material.

*Avena* L. spp.	Ploidy	Origin	Quantity
*A. sativa* L.	2 n = 42	Russia, Brazil, Germany, Belarus, USA	23
*A. byzantina* Coch.	2 n = 42	Finland, Ukraine, Great Britain, Portugal, USA, Kazakhstan, Germany, Brazil, Ethiopia, Russia	20
*A. abyssinica* Hochst.	2 n = 28	Ethiopia	3
*A. strigosa* Schreb.	2 n = 14	Brazil	3

## Data Availability

Not applicable.

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
