# Peer review of "Evaluating Germplasm of Cultivated Oat Species from the VIR Collection under the Russian Northwest Conditions"

_plants, 2022, doi:10.3390/plants11233280_

Round 1
Reviewer 1 Report
1. To add agronomic description of the interspecies hybrids which could make the manuscript more valuable to the readers.
2. Clearly write the materials and method, especially the field design including how many data points were collected for each line and how the data was treated if the design is augmented.
3. More explanation for the controversy correlation between starch and protein in 2014 and 2015.
4. Abstract needs to be more accurate and concise.
5. Conclusion needs to be more specific, not something like high beta-glucan for food and low beta-glucan for feed.
6. Carefully check spellings, The word “ standard” in table one may be “standard”?
Author Response
Thank you very much for all your comments and recommendations. We took them all into consideration and try to rewrite manuscript according to your recommendations, making edits to the text of the article.
Comments and suggestions for authors (1 reviewer)
Recommendation: To add agronomic description of the interspecies hybrids which could make the manuscript more valuable to the readers.
Answer: We supplemented by the available results with agro-technical characteristics of all samples, including hybrids, made as a result of field experiment.
Recommendation: Clearly write the materials and method, especially the field design including how many data points were collected for each line and how the data was treated if the design is augmented.
Answer: The material for the agro-technical indicators assessment is selected from five points of each plot. Points are located on the edges and in the center of the plot, at least 10 plants from the plot. Measurements of the oat’s stem length and the spike length are carried out using measuring rulers, 1.2 m and 50 cm long, respectively. The grain number of an individual spike and the number of spikelets are measured by simply counting. The weight of 1000 oat grains and yield per square meter are evaluated by lab balance (Sartorius L 610 D). Based on the measurement results, the average value and the range of the indicator’s variability for each oat sample are calculated.
Recommendation: More explanation for the controversy correlation between starch and protein in 2014 and 2015.
Answer: According to literature data, with a lack of moisture at elevated temperatures and a high nitrogen content in the soil, the metabolism in plants shifts towards increased synthesis of protein substances. With the reverse orientation of these factors, the processes of synthesis of carbohydrates and starch in seeds will prevail in plants [Pleshkov, B.P. Biochemistry of agricultural plants, 5th ed.; Agropromizdat: Moscow, Russia, 1987; p. 486 (in Russ.).]. This fact is confirmed by our correlation data for dry 2014 and wet 2015.
Recommendation: Abstract needs to be more accurate and concise.
Answer: We agree with the remark and have written the abstract more briefly and clearly.
Recommendation: Conclusion needs to be more specific, not something like high beta-glucan for food and low beta-glucan for feed.
Answer: The conclusions were corrected.
Recommendation: Carefully check spellings, The word “ standard” in table one may be “standard”?
Answer: The word "standard" has been corrected.

Reviewer 2 Report
Quite a bit is discussed about the different species but for several of the species the number of samples tested was relative small with only 3 entries. It would have been better to test larger numbers for these two species.
The entries tested was a relative small group and included only 49 of the 14,000 entries in the collection. A larger number would be better. It is apparent even with the small number that there is a lot of diversity for these traits in their collection.
It should be pointed out that these quality traits are important but there are other factors that are also important such as yield, disease resistance, maturity, lodging resistance, winter hardiness, etc.
Probably should mention whether the samples came from a spring or winter growing season - I am assuming that it was a spring growing season. Winter sown oats are grown in some countries.
Table 2. is filled with only 0.000 so I am unsure of the value of this table to the manuscript.
Probably the most useful data in the paper is in Supplementary materials Table 1 & 2.
They do mention some of the higher protein lines tested as compared to the standard variety "Privet"
They do identify individual accessions with potential to be used in a breeding program with different values for the traits measured.
I do like the inclusion of the summary of data for all entries in Table 3 and believe that it should be published as part of the paper.
Page 1 Line 10 - Oats are not the most important crop. But they are an important crop.
Page 2 Line 49 - "numerous studies confirm" rather than "than numerous studies are confirmed"
Page 2 Line 67 - "breeders work" rather than "in work breeders"
Page 4 Line 145-147 - low variability of A. strigosa and A. abyssinica may have been caused by relative low number of samples tested (only 3 tested for these species)
Author Response
Thank you very much for all your comments and recommendations. We took them all into consideration and try to rewrite manuscript according to your recommendations, making edits to the text of the article.
Comments and suggestions for authors (2 reviewer)
Recommendation: Quite a bit is discussed about the different species but for several of the species the number of samples tested was relative small with only 3 entries. It would have been better to test larger numbers for these two species.
Answer: We agree with the remark. However, for samples that have recently entered the collection, the stage of accumulation of a sufficient number of seeds is first carried out so that it is possible to deposit the sample for storage, and then, when a sufficient number of seeds is reached, the samples are studied in the methodological departments of the Institute.
Recommendation: The entries tested was a relative small group and included only 49 of the 14,000 entries in the collection. A larger number would be better. It is apparent even with the small number that there is a lot of diversity for these traits in their collection.
Answer: Thank you so much for your comment. It fully reflects the complexity of working with collectible material. The study of collection samples is carried out in stages, the collection is grown in open ground conditions and under the influence of external factors, the yield does not always allow for the inclusion of samples in the study with the involvement of methodological departments.
Recommendation: It should be pointed out that these quality traits are important but there are other factors that are also important such as yield, disease resistance, maturity, lodging resistance, winter hardiness, etc.
Answer: Thank you for your comment; we have supplemented the basic indicators of nutritional value with agro-technical indicators: plant height, spikelet length, number of spikelets, number of grains, weight of 1000 grains, and seed productivity (yield).
Recommendation: Probably should mention whether the samples came from a spring or winter growing season - I am assuming that it was a spring growing season. Winter sown oats are grown in some countries.
Answer: Samples of spring oats resistant to lodging were taken for study.
Recommendation: Table 2. is filled with only 0.000 so I am unsure of the value of this table to the manuscript.
Answer: We have moved this table to the Appendix.
Recommendation: Probably the most useful data in the paper is in Supplementary materials Table 1 & 2. They do mention some of the higher protein lines tested as compared to the standard variety "Privet". They do identify individual accessions with potential to be used in a breeding program with different values for the traits measured.
Answer: Yes, in Table 3 we have identified individual accessions with various useful traits.
Recommendation: I do like the inclusion of the summary of data for all entries in Table 3 and believe that it should be published as part of the paper.
Answer: Thank you very much for the comment. We have included Table 3 in the Appendix to the article
Recommendation: Page 1 Line 10 - Oats are not the most important crop. But they are an important crop.
Answer: We changed it to a phrase: "Oat is one of the most widespread and important grain crops"
Recommendation: Page 2 Line 49 - "numerous studies confirm" rather than "than numerous studies are confirmed"
Answer: We have changed the sentence "Numerous studies have confirmed the beneficial role of β-glucans on insulin resistance, hypertension, obesity, and dyslipidemia" to the next one: "Oats food products are beneficial for people with insulin resistance forms of diabetes, hypertension, obesity, and dyslipidemia".
Recommendation: Page 2 Line 67 - "breeders work" rather than "in work breeders"
Answer: We have changed this phrase in the next one: "They are offered for deeper investigation and subsequently usage in different breeding programs".
Recommendation: Page 4 Line 145-147 - low variability of A. strigosa and A. abyssinica may have been caused by relative low number of samples tested (only 3 tested for these species)
Answer: We agree with the remark. Unfortunately, the selection does not always turn out to be aligned with samples that have recently entered the collection. However, even a small number of samples of A. strigosa and A. abyssinica can reveal the biochemical and agro-technical features of these species. In addition, the field experiment is highly dependent on the climatic features of each year.

Round 2
Reviewer 1 Report
The revised version is better. Readers could extract some useful information from the manuscript such as the variations of the agronomic and quality traits as well as the responses of the germplasm lines in different growing environments. Those information will help potential researchers special breeders to use the germplasm resources from the center.